



# Quantifying the large-scale electrification equilibrium effects in dust storms using field observations at Qingtu Lake Observatory

**Huan Zhang[1], Xiaojing Zheng[1,2,*]**

[1]Key Laboratory of Mechanics on Disaster and Environment in Western China, The Ministry of Education of China, Department of Mechanics, Lanzhou University, Lanzhou 730000, China

[2]Research Center for Applied Mechanics, School of Mechano-Electronic Engineering, Xidian University, Xi'an 710071, China

*Correspondence to*: Xiaojing Zheng (xjzheng@xidian.edu.cn)



**Abstract**

Dust/sand electrification, which is a ubiquitous phenomenon in dust events, has a potentially dramatic effect on dust/sand lifting and transport processes. However, the effect of such electrification is still largely unclear, mainly due to its complexity and sparse observations. Here, we conducted an extensive observational analysis involving mild and severe dust storms with minimum visibility, ranging from ~0.09 to 0.93 km, to assess the electrical properties of airborne dust particles in dust storms. The space charge density has been estimated indirectly based on Gauss's law. Using the wavelet coherence analysis that is a method for evaluating the correlations between two non-stationary time series in the time-frequency domain, we found that the space charge density and dust concentration were significantly correlated over the 10 min timescales that is on the order of the typical integral time scale of atmospheric turbulence. We further presented a simple linear regression (SLR) model to quantify such large timescale correlations and found that there was a significant linear relationship between space charge density and dust concentration at given ambient temperature and relative humidity (RH), suggesting that the estimated mean charge-to-mass ratio of dust particles was expected to remain constant (termed as the equilibrium value $\mu^*$). In addition, the influences of ambient temperature and RH on $\mu^*$ were evaluated by a multiple linear regression (MLR) model, showing that the $\mu^*$ is nonlinearly related to environmental factors. The present study provides observational evidence for the environmental-dependent electrification



equilibrium effects in dust storms. This finding may reduce challenges in future quantifications of dust electrification, as it is possible to exclude effects, such as the

particles' collisional dynamics, on dust electrification.

## 1 Introduction

Granular materials in dust events such as blowing sand, dust devils, and dust storms, are frequently brought into contact or collision with each other, accumulating large amounts of electrical charge on their surfaces (Kamra, 1972; Rudge, 1913; Schmidt et al.,

1998). The strong electrostatic forces exerted on dust particles, which are comparable to gravitational force, could considerably affect the motion of particles and facilitate the lifting of particles from the ground (Esposito et al., 2016; Harper et al., 2017; Kok and Renno, 2008; Schmidt et al., 1998; Zheng et al., 2003). The first mutual coupling numerical saltation model (that is, wind flow, particle motion, and electrostatic forces interacting

with each other) developed by Zheng et al. (2003) showed that vertically down-pointing electrostatic forces strongly lowered particle trajectories (and conversely, upward-pointing forces elevated particle trajectories). Additionally, recent observations have demonstrated that electrostatic forces can enhance dust concentration by up to a factor of 10 when the intensity of the electric field (E-field) exceeds a certain threshold value

(Esposito et al., 2016). In addition to affecting dust transport, the E-fields created by charged dust may contribute to detection and early warning of severe dust events. For example, Aizawa et al. (2016) and Zhang et al. (2017) have documented that the



substantial enhancement in the intensity of E- fields occurs prior to the arrival of the dust clouds (fronts). Therefore, there is the considerable impetus to investigate electrical

effects in dust events.

Measurements of electrical effects in dust events (especially dust storms) have been made for over 100 years, dating back to 1913 with the finding that the atmospheric vertical E-fields increased by 1-3 orders of magnitude and reversed their direction during dust storms (Rudge, 1913; Schmidt et al., 1998; Williams et al., 2009). However, owing to

the difficulty in monitoring random dust events, only several tens of events are recorded in the literature (see Table 3 in Zheng, 2013, for details). Previous studies are mainly concerned with the vertical profile of E-fields in the near-ground region and their influence on dust transport (e.g., Esposito et al., 2016; Kamra, 1972; Rudge, 1913; Yair et al., 2016). The measurements showed that the vertical E-fields decrease with increasing

height in the sub-meter region (e.g., Schmidt et al., 1998), but exhibit multi-layer properties in the region of up to 30 m height (Zhang et al., 2017). Though measurements of E-fields in dust events are numerous, the measurements associated with the electrical properties of dust particles are relatively sparse. So far, only the charge-to-mass ratio of saltating particles (in the range of <10 cm height) has been measured using a Faraday

cage, showing that its magnitude is on the order of ~60 µC kg$^{-1}$ (Bo et al., 2014; Schmidt et al., 1998). To better understand the electrical properties of airborne dust particles, further measurements are therefore required.



Furthermore, theoretical work and laboratory experiments have shown that the net electrical charge on granular particles increases with increasing number of collisions/contacts and correlates with the particles' kinetic energy (Apodaca et al., 2010; Harper and Dufek, 2016; Matsuyama and Yamamoto, 1995; Zhang et al., 2013) until particles acquired a certain amount of charge (termed as equilibrium charge). The magnitude of the eventual equilibrium charge on particles is found to be independent of the particles' collisional dynamics and therefore will reduce the difficulties in the quantification of particle electrification. However, whether such an electrification equilibrium exists under natural circumstances, especially in dust storms, is unclear and needs to be verified. The ratio of space charge density to the dust mass concentration (called mean charge-to-mass ratio of dust particles) rather than charge on the individual particles is generally used to quantify the electrical properties of dusty phenomena. Additionally, preliminary research (Esposito et al., 2016; Merrison, 2012; Xie and Han, 2012; Zhang et al., 2017; Zheng et al., 2014) suggests that environmental factors such as the ambient temperature and RH could considerably affect the intensity of E-fields. It is still unknown whether environmental factors have an impact on the properties of dust electrification, and especially electrification equilibrium effects in a dust storm.

In this study, we build on a set of field observations through an extensive statistical analysis to assess the mean scaled charge-to-mass ratio of airborne dust particles $\mu^*$ (defined in Sect. 2.2) in dust storms and to untangle the influences of environmental



factors (i.e., ambient temperature and RH) on the $\mu^*$. Similar to the indirect method for

determining the space charge density of thunderstorms, clouds and aerosol layers in the

atmosphere (e.g., MacGorman and Rust, 1998; Nicoll, 2012; Stolzenburg and Marshall,

1994; Zhou and Tinsley, 2007), we estimated the multi-meter averaged space charge

density based on Gauss's law, in which the mean space charge density is proportional to

the divergence of the E-fields, as described in detail in Sect. 2.2. We evaluated correlations

between the space charge density and the dust concentration in time and frequency

space through wavelet coherence analysis. We then used an SLR model to assess whether

the electrification equilibrium effects exist in dust storms. Finally, we developed an MLR

model to evaluate the effects of ambient temperature and RH on the equilibrium charge.

## 2 Methods

### 2.1 Description of the field site

We conducted field observations in a flat-bottomed dry lakebed of the Qingtu Lake

(approximately 103°40′03′′ E, 39°12′27′′ N), approximately 90 km northeast of Minqin,

Gansu, China (Fig. 1a), over the period from 21 March to 2 June 2017. The area was

selected since it lies within a dusty belt in the Hexi Corridor (Wang et al., 2018), which is

mainly affected by the Mongolian cyclones (and probably by the cold downdrafts from

thunderstorms/squall) during the observational period and is therefore frequently

subjected to dust events (Shao, 2000; Williams et al., 2009). At the Qingtu Lake

Observational Array (QLOA) site, there is a prevailing wind direction, along with a straight



line having a mean angle of ~247.5° with respect to the North (approximately 74.6 %

frequency), as shown in Fig. S1 in the Supplement. Measurements of size distribution of

saltating particles shows that the dust events occurring in the QLOA site have a very

similar particle size distribution (Fig. S2). The QLOA site consists of 21 observation towers

distributed in two mutually perpendicular directions (the prevailing wind direction and

the spanwise direction). Among these towers, the main tower with a 32 m height could

be used to measure the vertical E-field gradients, and the remaining 20 towers with 5 m

height could be designed to determine the streamwise and spanwise gradients of E-fields

(Fig. 1b).

**2.2 Measurements**

In dust storms, space charge density $\rho$ can be expressed as the product of the mean

charge-to-mass ratio of dust particles $\mu$ and the total dust mass concentration $M_{10}/\lambda$,

where $M_{10}$ and $\lambda$ denote the $PM_{10}$ mass concentration and the $PM_{10}$ mass fraction,

respectively. As mentioned in Sect. 2.1, the size distribution of airborne dust particles

varies slightly from event to event. Consequently, the scaled mean charge-to-mass ratio

$\mu^*$, which is a common measure of the charge-to-mass ratio of dust particles, can be

defined as


$$\mu^* \equiv \frac{\rho}{M_{10}} \qquad (1)$$



where $\mu^* = \mu/\lambda$ ($\lambda$ is assumed to be a constant) is equal to the mean charge-to-mass ratio

divided by the $PM_{10}$ mass fraction. From this definition, $\mu^*$ can be determined once the

space charge density and $PM_{10}$ mass concentration have been determined. By adopting

Standard International units, the units of $\mu^*$ and $\mu$ are C kg$^{-1}$, the unit of $\rho$ is C m$^{-3}$, the

unit of $PM_{10}$ concentration $M_{10}$ is kg m$^{-3}$, and the $PM_{10}$ mass fraction $\lambda$ has a dimensional

unit in Eq. (1).

In the 2017 field campaign, the indirect method mentioned in the introduction was

used to estimate the $\mu^*$ at a site 5 m height above the ground (i.e., S9). A total of 11

measurement sites (S1-S11 distributed on 7 observation towers) were arranged to

estimate the space charge density at S9, as shown in Figs. 1b and 1c. In theory, the space

charge density ρ at S9, which can directly reflect the polarity of the dust cloud at this

height, is related to the corresponding divergence of the E-fields $\nabla \cdot \mathbf{E}$ by Gauss's Law

(Pollack and Stump, 2002)

$$\rho = \varepsilon_0 \frac{\partial E_i}{\partial x_i} \tag{2}$$

where $\varepsilon_0 = 8.854 \times 10^{-12}$ C$^2$ N$^{-1}$ m$^{-2}$ is the permittivity constant in a vacuum; suffix $i$ is

summed from 1 to 3 by making use of Einstein summation convention. In Eq. (2), to

determine the partial derivatives of E-fields $E_i$ with respect to $x_i$ at S9, E-fields sensors

called vibrating-reed electric field mill (VREFM), which were developed by Lanzhou



University (Zheng, 2013), were installed in the streamwise, spanwise, and wall-normal monitoring networks (Fig. 1c). The VREFM functions were based on measuring the induced electrical charge and were calibrated by a large parallel-plate E-field calibrator (one-meter square plates). More detailed information of the VREFM is given in our previous works (see Zhang et al., 2017; Zheng, 2013). The partial derivatives of E-fields were estimated from the interpolation-based numerical method and will be shown to be both positive and negative (see Fig. S4 in the Supplement).

The $PM_{10}$ mass concentration at S9 was measured by a DustTrak II Aerosol Monitor (Model 8530EP, TSI Incorporated). Additionally, to explore the effects of environmental factors on $\mu^*$, additional instruments installed at S12 include: sonic anemometer (CSAT3B, Campbell Scientific), measuring three-dimensional (3D) wind velocity; a sand particle counter (SPC-91, Niigata Electric Co., Ltd.), measuring saltating sand particle number density in the range of ~30-490 μm with 32 bins; a temperature-humidity sensor (Model 41003, R. M. Young Company), measuring ambient temperature and RH; and visibility sensor (Model 6000, Belfort Instrument), measuring visibility ranging from 5 m to 10 km with ±10 % accuracy (Fig. S3 in the Supplement). Before performing field measurements, all instruments were carefully calibrated in the laboratory. The VREFM sensors were also calibrated at QLOA site by comparing its output to a higher accuracy atmospheric E-field mill (see Fig. S7 in the Supplement). To achieve the best possible instrument accuracy, we performed re-calibration for VREFM sensors and periodic cleaning for Aerosol Monitor



8530EP twice a month during the observational period. All instruments were monitored

continuously and simultaneously with a sampling frequency of 1 Hz (except for the

CSAT3B which had a sampling frequency of 50 Hz).

### 2.3 Wavelet coherence analysis

For two stationary time series, the ensemble-mean values are equal to the time-

mean values; thus, the cross-correlation function and cross-spectral density function are

respectively used to measure the joint statistical properties in the time and frequency

domain (Bendat and Piersol, 2011). In practice, however, the geophysical time series is

generally non-stationary (see Fig. 2) and therefore consists of time-varying statistical

properties (Grinsted et al., 2004; Holman et al., 2011). In this study, wavelet coherence

analysis is used to assess the correlations between the space charge density ρ and the

$PM_{10}$ dust concentration $M_{10}$ in the time-frequency domain. For two time series,

X and Y, the squared wavelet coherence is defined as (Grinsted et al., 2004)

$$R^2(n,s) = \frac{|\langle s^{-1}W^{XY}(n,s)\rangle|^2}{|\langle s^{-1}W^X(n,s)\rangle|^2|\langle s^{-1}W^Y(n,s)\rangle|^2} \tag{3}$$

where the brackets $\langle\ \rangle$ are the smoothing operator in time and scale; $W^{XY} = W^X W^{Y*}$

is the cross wavelet transform; and superscript $^*$ is the complex conjugation. The Morlet

wavelet (with $\omega_0$ =6) is used to perform the continuous wavelet transforms $W^X$

and $W^Y$. The squared wavelet coherence $R^2(n,s)$ ranges from 0 to 1 and can be



thought of as a localized correlation coefficient between two time series in time and frequency space (Grinsted et al., 2004; Holman et al., 2011; Wang et al., 2017). A Matlab

toolbox for wavelet coherence analysis can be found at http://noc.ac.uk/using-science/crosswavelet-wavelet-coherence (Grinsted et al., 2004; Wang et al., 2017).

**2.4 Regression analysis**

An SLR model was used to explore whether the electrification equilibrium effects exist in dust storms. Theoretically, there is an electrification equilibrium effect (i.e., $\mu^*$ is

constant) if ρ and $M_{10}$ are linearly correlated at a given ambient temperature and RH (and obviously, nonlinear relationship between them means time-varying $\mu^*$). Since previous studies (Esposito et al., 2016; Harper and Dufek, 2016; Merrison, 2012; Wiles et al., 2007; Xie and Han, 2012; Zhang et al., 2017; Zheng et al., 2014) suggested that environmental factors such as ambient temperature and RH could considerably affect the intensity of E-

fields, SLR in this study were performed at a set of given temperature and RH intervals of length 2 °C and 2 %, respectively. Although theoretical consideration suggests that the intercept of the linear model (Eq. 1) should be zero, it is also present due to the inaccuracy of measurements and the requirement of statistical inference procedures for linear models. Thus, the SLR model has the form


$$\rho = a_0 + a_1 M_{10} + \varepsilon_1 \tag{4}$$


where coefficient $a_0$ is the fitted intercept, the fitted slope $a_1$ is equal to $\mu^*$, and $\varepsilon_1$ is a disturbance term. We use the ordinary least squares method to estimate the

coefficients of the SLR model (Eq. 4). The F-test is used for testing the SLR model.

Previous studies have shown that charge transfer processes are nonlinearly related to the ambient temperature and RH (Gu et al., 2013; Lacks and Sankaran, 2011; McCarty and Whitesides, 2008; Wei and Gu, 2015; Zheng et al., 2014). We thus use the following MLR to evaluate the effects of ambient temperature and RH on $\mu^*$:


$$\mu^* = b_0 + b_1 T_a + b_2 RH + b_3 T_a^2 + b_4 RH^2 + b_5 T_a RH + \varepsilon_2 \tag{5}$$

where $T_a$ and $RH$ are the ambient temperature and $RH$, respectively; $b_i$ ($i$=0, 1,. . ., 5) is the fitted coefficients; and $\varepsilon_2$ is a disturbance term. The multiple regression

analyses were undertaken with R (v.3.4.1).

**3 Results**

**3.1 Electrification equilibrium effects over large timescales**

Fortunately, we have successfully recorded 10 dust storms lasting a total of ~66 h, which is sufficient to perform a series of reliable statistical analyses (Table 1). As an

example, Fig. 2 shows the complete data series recorded during a severe dust storm (Movie S1) with the maximum streamwise E-field intensity of up to ~181 kV m$^{-1}$ (much less than the dielectric strength of air), suggesting that dust particles are highly electrified.





As shown in Fig. 3, on timescales of less than 2 min, there is very low wavelet coherence between $\rho$ and $M_{10}$ throughout the whole period of dust storms, suggesting

that the high-frequency fluctuation of dust concentration (probably due to turbulence) is not associated with the space charge density. In contrast, $\rho$ and $M_{10}$ are significantly in-phase correlated on timescales longer than ~10 min (with coherence power of up to ~0.9-1.0). Actually, the integral time scale of atmospheric turbulence is on the order of ~10 min (Durán et al., 2011). The wind variations over time scales smaller than ~10 min are

attributed to turbulence, while variations over time scales larger than ~10 min are attributed to meteorological effects. In general, the aeolian transport and wind strength are highly correlated over time scales larger than ~10 min. Since the fine dust particles often follow the wind strictly, the large timescale strong correlation between $\rho$ and $M_{10}$ are certainly reasonable where the effects of turbulent fluctuations are excluded. In

addition, the behavior of such significant large-timescale correlation (as indicated by the yellow area) is quite different from the observed dust storms (Fig. 3). That is, over 10 min timescales (period larger than 10 min in Fig. 3), the significant correlation is present in the whole time of Fig. 3 b, e, f, g, and j but is locally distributed in the time of Fig. 3 a, c, d, h, and i.

To quantify the strong large timescale correlations between $\rho$ and $M_{10}$, we performed SLR analysis between the 10 min moving average (See Fig. S5 in the Supplement) of the $\rho$ and $M_{10}$ time series, where the fitted linear regression slope is equal



to the $\mu^*$. The SLR analysis was performed for a set of given temperature and RH intervals

(within 2 °C and 2 %). As shown in Fig. 4, there is a significant linear relationship between

ρ and $M_{10}$ at a given ambient temperature and RH (with median $R^2$ of ~0.71-0.98 and

p<0.01, see Fig. S6 and Table S1 in the Supplement), suggesting that $\mu^*$ is nearly constant

during a period that ambient temperature and RH are fixed. The long period constant $\mu^*$

implies that electrification equilibrium has been established (on average) where the rates

of gain and loss of electrical charge are equal. $\mu^*$ is significantly influenced by

environmental factors but independent of the particles' collisional dynamics and wind

speed. For example, in Fig. 4, $\mu^*$ (i.e., slope) varies from 0.264 to 0.421 C kg$^{-1}$ for

different ambient conditions but is constant at a given ambient condition, even though

the wind speed change dramatically.

**3.2 Temperature and RH dependence of $\mu^*$**

As shown in Fig. 5a, we find that $\mu^*$ varies widely from 0 to 3 C kg$^{-1}$ for different

ambient temperatures and RHs, consistent with the previous finding of the significant

environmental dependence of granular electrification (Lacks and Sankaran, 2011;

McCarty and Whitesides, 2008). According to the SLR analysis between ρ and $M_{10}$, we

used an MLR model to quantify the ambient temperature and RH dependence of $\mu^*$. The

results of estimating coefficients, F-test for testing the model, and T-test for testing

coefficients are shown in the Table S2. It is seen from Fig. 5a that the MLR model is in

good agreement with the measurement-based data ($R^2$= ~0.71, p<0.001). Clearly, the



effects of ambient temperature and RH on the saturation $\mu^*$ are coupled and behave

quite differently at different conditions (Figs. 5b and c), as predicted by the MLR model.

For example, $\mu^*$ increases (decreases) with increasing RH at $T_a$ = 27.5 °C (5.5 °C), while

it decreases first and then increases at $T_a$ =16.5 °C, as shown in Fig. 5b. For various RH

(8.5 %, 25.5 %, and 42.5 %), $\mu^*$ showed a similar pattern with increasing temperature:

$\mu^*$ first decreased and then exhibited an upward trend, as shown in Fig. 5c.

## 4 Discussion

**4.1 Methods for estimating space charge density**

In granular materials, the electrification of dust particles is generally evaluated by

the surface charge density and the charge-to-mass ratio (Merrison, 2012; Schmidt et al.,

1998). The former is defined as the charge on a particle divided by its surface area, and

the latter is defined as the charge on a particle divided by its mass. For laboratory

experiments, the mean charge-to-mass ratio and even the surface charge density

distribution on an individual particle can be accurately determined using optical

microscopy such as particle image velocimetry (PIV, see Waitukaitis and Jaeger, 2013;

Waitukaitis et al., 2014) and Kelvin force microscopy (KFM, see Baytekin et al., 2011b).

However, such methods based on optical microscopy are difficult to use in field

observations due to very complex environmental conditions (Harrison et al., 2016;

Merrison, 2012). In this study, as in the method investigating the electrical effects of

thunderstorms, clouds, and aerosol layers in the atmosphere (e.g., MacGorman and Rust,



1998; Nicoll, 2012; Stolzenburg and Marshall, 1994; Zhou and Tinsley, 2007), the mean space charge density, which is the quantity of charge per unit volume, was determined indirectly. According to Gauss's law, the total space charge density is directly proportional to the divergence of the E-fields. The VREFMs spacing is respectively ~1.6, 5, and 10 m in the vertical, spanwise, and streamwise directions owing to the rapid variation of E-fields along the vertical direction and slow variation along the spanwise and streamwise directions (see Fig. S4 in the Supplement), in order to eliminate the disturbances from other VREFMs and the observation tower, and thus the calculated space charge density is actually a measure of the mean charge of dust particles in a multi-cubic meter volume. As shown in Fig. 4, the calculated space charge density is ~$10^{-8}$-$10^{-7}$ C m$^{-3}$ corresponding to ~$0.624\times10^{5}$-$10^{6}$ el m$^{-3}$ (el denotes elementary charge), in accordance with the order of $0.7\times10^{6}$ el m$^{-3}$ in a New Mexico dust devil (Crozier, 1964). It is worth noting that E-fields as high as ~181 kV m$^{-1}$ are rarely found at a few meters height in the literature (except within the dust devils where horizontal E-fields exceeded 100 kV m$^{-1}$, see Jackson and Farrell, 2006), because the charging processes between granular materials are very sensitive to the properties and size distribution of grains (Houghton et al., 2013). In our study, the saline-alkali soil at the QLOA site may facilitate the charge separation of dust particles, which may lead to high E-field intensity in dust storms.

**4.2 The physical mechanisms for electrification equilibrium**

Previous granular studies (e.g., Harper and Dufek, 2016; Matsuyama and Yamamoto,



1995) have shown that the individual particles in granular flows were able to acquire a certain amount of saturation or equilibrium charge, where no further charge transfer occurs in spite of subsequent impacts between particles. The physical mechanisms for electrification saturation of individual particles can be attributed to the dielectric breakdown of air (Harper and Dufek, 2016; Matsuyama and Yamamoto, 1995; McCarty et al., 2007) and a repelling E-field in the charge transfer direction (Castle and Schein, 1995). In the present study, the large-scale electrification equilibrium effects widely exist in dust storms (Figs. 3 and 4). However, in dust storms, we propose that such electrification equilibrium of a large-scale system (averaged over multi-cubic meter volume and 10 min) is a dynamic equilibrium rather than the saturation of individual particles. In this case, the charges on dust particles transfer between the large-scale systems at an equal rate, meaning there is no net charge exchange. Charge transfer between individual dust particles may in fact occur, but to such an extent that we cannot observe the changes in $\mu^*$ of the large-scale system under certain ambient condition. It should be emphasized that the concept of large-scale electrification equilibrium is only applied to the dust storms under certain ambient condition; that is, $\mu^*$ is constant with varying particles' dynamics at given temperature and RH. Once ambient temperature or RH is changed, the large-scale system will reach a new electrification equilibrium. Consequently, such equilibrium can be termed environmental-dependent equilibrium effects. The equilibrium state is occasionally broken down (characterized by a weak large-scale





correlation between space charge density and dust concentration) in several dust storms (e.g., Figs. 3c and d, as well as little $R^2$ in Fig. S6) and the reason for this is unclear. Further

research is needed to explore the mechanisms leading to the occasional absence of significant large-timescale correlations.

In addition, the equilibrium value ($\mu^*$) of the large-scale system was found to be strongly influenced by RH and ambient temperature in dust storms during our field observations. While water is not necessary for contact electrification (Baytekin et al.,

2011a), a variety of studies indicated that such charge separation was strongly dependent on the RH (Esposito et al., 2016; McCarty and Whitesides, 2008; Xie and Han, 2012; Zhang et al., 2017). The proposed reasons for this are twofold: On one hand, the presence of adsorbed water could increase surface conductivity and facilitate the ion or electron transfer (McCarty and Whitesides, 2008); On the other hand, OH⁻ ions in adsorbed

surface water could also act as charge carrier (Gu et al., 2013; Lacks and Sankaran, 2011; McCarty and Whitesides, 2008). We also found that $\mu^*$ was strongly affected by the ambient temperature. This is consistent with other reports, which showed that the dielectric constant and conductivity of the adsorbed water were significantly linked to the ambient temperature (Gu et al., 2013; Lacks and Sankaran, 2011; Wei and Gu, 2015). As

shown in Figs. 5b and 5c, the $\mu^*$ is nonlinearly related to ambient temperature and RH. This result has also been verified by other studies (Xie and Han, 2012; Zheng et al., 2014).

**4.3 Implications for quantifying electrostatics in dust storms**





Dust electrification is very complex due to particle-particle interactions (a non-equilibrium charging process involving a wide range of length and time scales) and
particle-turbulence interactions (particle inertia varying over several orders of magnitude at high Reynolds-number flow in dust storms). To quantitatively predict the charge transfer processes between dust particle collisions/contacts, a large number of theoretical models have been developed based on different physical mechanisms, such as asymmetric contact (Hu et al., 2012; Kok and Lacks, 2009; Kok and Renno, 2008),
polarization by external E-fields (Pähtz et al., 2012), statistical variations of material properties (Apodaca et al., 2010) and shift of aqueous ions (Gu et al., 2013). Moreover, the charge carriers among them could be electrons (Kok and Lacks, 2009), ions (McCarty and Whitesides, 2008), trapped holes (Hu et al., 2012), and bits of nanoscopic material (Apodaca et al., 2010), depending on the material properties and condition involved
(Lacks and Sankaran, 2011; Wei and Gu, 2015). These models mainly focus on explaining size-dependent charging (that is, smaller particles tend to acquire a net negative charge during collisions with larger particles) of dust particles without considering the environmental factors, which have been found to considerably affect dust electrification. Currently, the existing electrification models cannot explicitly account for the effects of
environmental factors RH and ambient temperature (Harrison et al., 2016; Lacks and Sankaran, 2011; Wei and Gu, 2015), and thus field observations are the primary way for exploring dust electrification.



Dust storms act as a natural experiment through which the granular electrification effects can be evaluated. We found that space charge density and dust concentration were significantly correlated over large timescales. This is clear observational evidence of the large-scale electrification equilibrium effects in dust storms. This finding has potentially important implications for exploring the role of electrical effects in dust events. First, the electrification equilibrium will be helpful in quantifying the electrical effects during dust storms. The fact that the charge transfer process depends strongly on particle dynamics has been previously demonstrated (Hu et al., 2012; Kok and Lacks, 2009; Wei and Gu, 2015). To quantify the particle charges (and therefore electrostatic forces) exactly, tracking the history of particle dynamics is required. However, it is too complex to be determined due to the polydisperse two-phase flow (dust storms) at very large Reynolds numbers of up to $Re_\tau \approx 4\times10^6$ (Wang and Zheng, 2016). Our results suggest that particle dynamics can be excluded reasonably well in future quantification of electrical effects. Second, the theoretical understanding of charging electrification is expected to be facilitated by the equilibrium effects. As mentioned previously, the underlying physical mechanism behind dust electrification is extremely complex because such electrification is a non-equilibrium phenomenon that involves a wide range of length and time scales (Lacks and Sankaran, 2011; Wei and Gu, 2015). The presence of electrification equilibrium in dust storms allows us to investigate the equilibrium mechanism, in contrast to the dynamic charging mechanism, which is believed to be straightforward.




## 5 Conclusions

One of the most important and difficult issues in investigating wind-erosion

phenomenon is the accurate quantification of the dust/sand electrification. As noted

previously, electrical effects can considerably influence the dust transport and lifting

processes. In this study, to evaluate the electrical properties of airborne dust particles in

dust storms, we performed continuous field measurements at QLOA site from 21 March

to 2 June 2017.

Overall, we have successfully monitored 10 complete dust storms lasting a total of

~66 h. According to a series of extensive analyses, we found that the average space charge

density and dust concentration were significantly correlated over 10 min timescales,

providing an observational evidence for the large-scale (over 10 min and multi-cubic

meter volume) electrification equilibrium effects in dust storms, even though such effects

are absent occasionally. In addition, the equilibrium scaled charge-to-mass ratio $\mu^*$ is

found to be independent of wind speed but is a nonlinear function of the ambient

temperature and RH. Our findings have potentially important implications for exploring

the role of electrical effects in dust events, especially in the aspect of the quantification

of electrical effects in dust storms. In further E-fields model of dust storms, the charge-

to-mass ratio of airborne dust particles can be considered as a constant if the ambient

temperature and RH do not change dramatically.



**Data availability**. The experimental data that support the findings of this study are available in Zenodo data repository, https://doi.org/10.5281/zenodo.1188752.


**Competing interests**. The authors declare that they have no conflict of interest.

**Acknowledgements**. We thank H.H. Gu, X.B. Li, A. Mei, G.W. Han, Y.R. Liang, G.H. Wang, H.Y. Liu and W. Zhu for providing useful assistance in the field observations, and thank

professor Y.H. Zhou for helpful comments and discussions. We acknowledge support from the National Natural Science Foundation of China (No. 11490553).

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


**Table 1.** Overview of the observed 10 dust storms. $\rho_{max}$, $\mu^{*}_{max}$, $E_{z,max}$, and $M_{10,max}$ denote the maximum value of estimated space charge density, scaled charge-to-mass ratio, intensity of vertical E-field, and $PM_{10}$ mass concentration during a dust storm; $T_a$ and RH denote the range of ambient temperature and relative humidity during a dust storm; $V_{b,min}$ denotes the minimum visibility during a dust storm.

| No. of Dust storms | Period (UTC+8) | $\rho_{max}$ (C m$^{-3}$) | $\mu^{*}_{max}$ (C kg$^{-1}$) | $E_{z,max}$ (kV m$^{-1}$) | $M_{10,max}$ ($\times 10^{-6}$ kg m$^{-3}$) | $T_a$ (°C) | RH (%) | $V_{b,min}$ (km) |
|---|---|---|---|---|---|---|---|---|
| 01 | 2017.04.17/10:00-19:00 | $0.43\times10^{-6}$ | 0.41 | 94.3 | 2.77 | 16.9-21.6 | 16.4-23.8 | 0.09 |
| 02 | 2017.04.18/15:00-18:00 | $0.71\times10^{-7}$ | 0.47 | 14.7 | 0.19 | 17.1-18.9 | 32.7-37.9 | 0.55 |
| 03 | 2017.04.18/22:00-2017.04.19/05:00 | $0.11\times10^{-6}$ | 0.92 | 23.8 | 0.36 | 9.7-15.1 | 12.7-43.1 | 0.41 |
| 04 | 2017.04.19/06:00-22:00 | $0.10\times10^{-6}$ | 0.87 | 23.9 | 0.25 | 8.8-17.2 | 8.1-47.4 | 0.78 |
| 05 | 2017.04.20/08:00-17:00 | $0.18\times10^{-6}$ | 0.49 | 43.4 | 0.72 | 6.4-13.8 | 10.9-22.1 | 0.34 |
| 06 | 2017.04.22/12:00-17:00 | $0.38\times10^{-6}$ | 2.15 | 63.3 | 0.17 | 21.2-23.7 | 4.2-6.2 | 0.57 |
| 07 | 2017.05.03/08:00-18:00 | $0.23\times10^{-6}$ | 0.89 | 40.6 | 0.99 | 10.5-16.8 | 7.7-53.1 | 0.24 |
| 08 | 2017.05.10/10:00-19:00 | $0.83\times10^{-7}$ | 1.1 | 28.2 | 0.08 | 22.3-26.5 | 7.1-10.2 | 0.93 |
| 09 | 2017.05.12/11:00-16:00 | $0.44\times10^{-6}$ | 2.68 | 71.2 | 0.18 | 28.1-31.2 | 3.4-6.4 | 0.49 |
| 10 | 2017.05.20/23:00-2017.05.21/01:00 | $0.27\times10^{-6}$ | 0.69 | 20.1 | 0.39 | 21.4-23.7 | 18.4-21.9 | 0.55 |






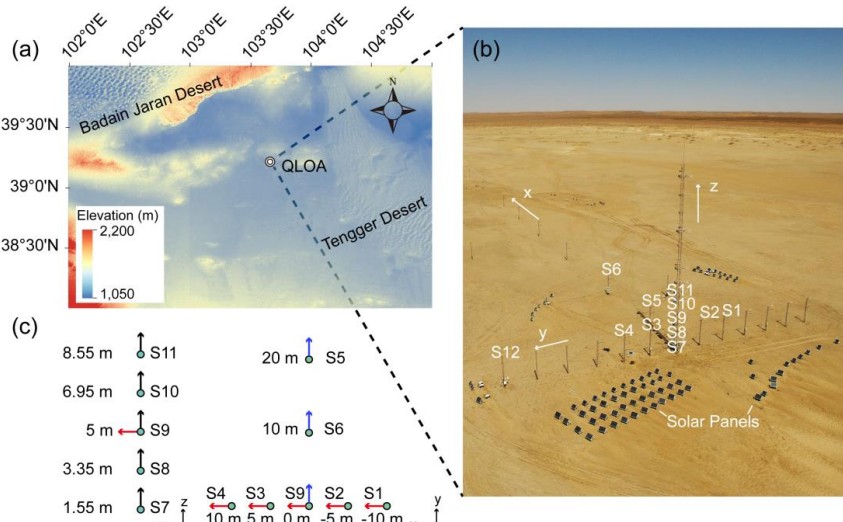

**Figure 1.** Location of the field site and the observational set-up. **(a)**: The QLOA site is located between the Badain Jaran Desert and the Tengger Desert (a dusty belt having high-frequency dust events). **(b)**: E-fields were measured at sites S1-S11; 3D wind velocity was measured at centrally-located S9. The ambient temperature, RH, saltating sand number density, and visibility were measured at S12. All of the instruments were powered by a solar panel system. The $x$, $y$, and $z$ axis are parallel to the prevailing wind direction, spanwise direction, and vertical direction, respectively. **(c)**: Layout of VREFMs (not to scale); that is, blue, red, and dark arrows correspond to the $x$, $y$, and $z$ component measurements of E-fields, respectively. It is worth noting that the $x$ and $y$ components of E-fields are generally non-zero because dust transport is non-uniform in the horizontal plane (Zheng, 2013).



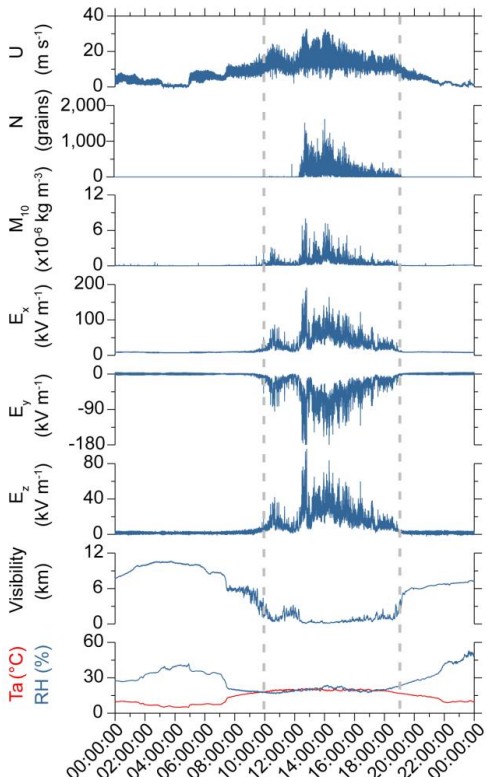

**Figure 2**. Data series recorded on 17 April 2017 (UTC+8). Top-to-bottom: streamwise wind speed, saltating sand number passing through the measurement area (2 mm in height and 25 mm in length) per second, $PM_{10}$ mass concentration, and 3D E-fields at S9; visibility, ambient temperature, and RH at S12. The E-fields $E_x$, $E_y$, and $E_z$ are positive if they point in the positive directions of $x$, $y$, $z$ axes depicted in Fig. 1. That is, $E_z$ and fair-weather atmospheric E-field are oppositely directed.





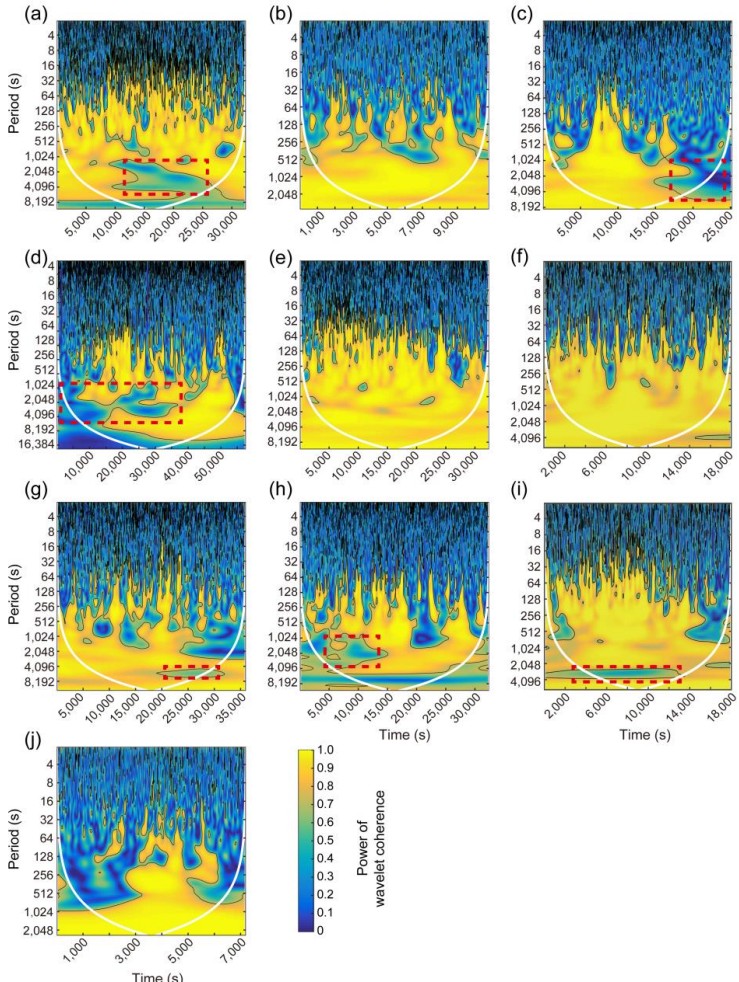


**Figure 3**. Coherence analyses between the space charge density and dust concentration.

**(a)-(j)**: Squared wavelet coherences, in turn, correspond to No. 1-10 dust storms

presented in Table 1. The 5 % significance level, estimated by Monte Carlo methods

(Grinsted et al., 2004), against red noise is shown as black contour lines. The cone of

influence where edge effects might distort the signal is shown as a white line. The relative

phase relationship between the two time series is shown in Figs. S8-S12. Dashed

rectangular boxes denote the destructions of large-scale electrification equilibrium at

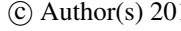



some time.

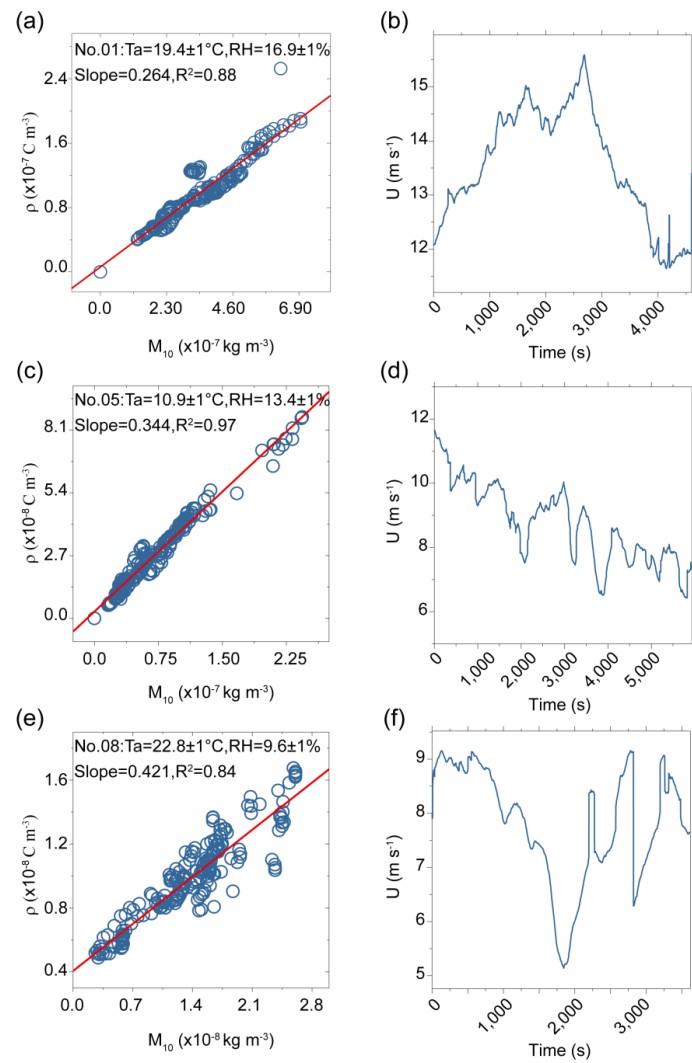

**Figure 4**. Significant linear relationships between the 10 min moving average of space charge density and dust concentration at the given ambient temperature and RH. **(a)**, **(c)**, and **(e)**: Circles denote the data obtained from observations, and straight lines denote the linear regression. The slopes corresponds to the scaled charge-to-mass ratios $\mu^*$. **(b)**, **(d)**, and **(f)**: Wind speed during the corresponding period of **(a)**, **(c)**, and **(e)**.




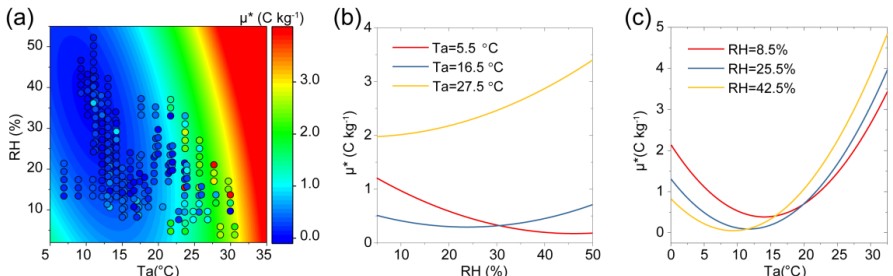

**Figure 5.** Dependence of the $\mu^*$ on environmental factors. **(a)**: Comparison of the MLR

model (contour) with measurement-based data (circle). **(b)**: The predicted $\mu^*$ as a

function of RH for different ambient temperatures. **(c)**: The predicted $\mu^*$ as a function of

ambient temperature for different RHs.