# Peer review of "Quantifying the large-scale electrification equilibrium effects in dust storms using field observations at Qingtu Lake Observatory"

_Atmospheric Chemistry and Physics, 2018_

## Referee Comment (RC1) · J. Merrison (Referee) · 13 Sep 2018

MS No.: acp-2018-293

Quantifying the large-scale electrification equilibrium effects in dust storms using field observations at Qingtu Lake Observatory Author(s): Huan Zhang and Xiaojing Zheng

This paper presents the results and analysis of a field campaign studying the electric field generated by a series of dust storms. This work is of high interest to those studying Aeolian processes, atmospheric electricity, aspects of planetology and especially the

electrification of aerosols and granular materials. This is one of only a few such studies and it appears to be thorough and in depth applying a range of experimental sensor techniques.

General comments Generally in this paper the distinction between sand and dust is unclear, an example is line 155 in which the size distribution of saltating particles (by definition these must be sand) are used to make arguments about dust events. By definition sand grains cannot be suspended and are transported at low altitude (typically <1m), cohesion effects for sand are typically negligable. Dust cannot saltate and it is unclear whether they can/do in fact collide while in suspension, it may be that if in contact dust grains will cohere (aggregate). It is my impression that this study focusses on atmospheric dust in which case this should be made clear.

Specific comments It is of some interest whether the field measurements made here are consistent with dust being typically electrified negatively, the authors might want to comment upon this.

The consecutive acquisition of charge (Line 73-80) through collisions and a so called equilibrium charge has been observed only in some laboratory studies (mostly using sand sized particles), it is not demonstrated in all experiments and is not generally accepted that electrification of dust in fact involves multiple collisions. Recent laboratory work implies that it is not (e.g. Alois et al. 2017, 2018). Similarly it seems that the work presented here is not in fact in agreement with a model based upon multiple collisions (e.g. line 259, 406). (Alois, S., Merrison, J., Iversen, J.J., Sesterhenn, J., (2017) Contact electrification in aerosolized monodispersed silica microspheres quantified using laser based velocimetry, Journal of Aerosol Science 106 1–10., Alois, S., Merrison, J., Iversen, J.J., Sesterhenn, J., (2018), Quantifying the contact electrification of aerosolized insulating particles, Powder Technology 332, 106–113)

As the authors point out in laboratory studies it appears that the charge concentration (per surface area) is a more useful physical parameter than charge to mass ratio ($\mu$),

e.g. line is it possible to derive such a parameter from these measurements?. Alternatively information of charge per dust particle might in this case also be valuable.

It is of great interest that the observations presented here show a dependence upon RH especially as stated by the authors that the composition of the dust (soil) might imply a sensitivity to surface moisture (line 309). As the authors also point out some studies demonstrate dependence upon RH and others do not (line 337-347). Recent work has also shown that electrification can occur at extremely low RH but that high RH may greatly enhance electrification for some materials. This appears to present a consistent picture (Alois 2018).

---

## Referee Comment (RC2) · Anonymous Referee #2 · 20 Sep 2018

This is a well-organized study of natural dust storm electrification, with novel analysis and new findings. The English has been meticulously prepared. Some improvements are in order pertaining to the physical interpretation and the real evidence for equilibrium effects. A number of substantive issues are worth addressing by the authors in the preparation of their final manuscript. These issues are followed by detailed edits/comments on the text.

Summary: Publish after appropriate revision

[Figure]

Substantive Issues:

(1) Physical origin of dust events

The physical/meteorological basis for the events with other extensive documentation in this work is not elaborated. Lines 110-111 suggest a role for straight line winds. Are the cold downdrafts from thunderstorms/squall lines important for these events, as was the case in Niger in a study by Williams et al. (Atmos. Res., 2009). (We are aware of earlier thunderstorm studies in the Lanzhou area of China by other investigators—S. Liu for example.)

(2) Physical hypotheses for "equilibrium effects"

First of all, the physical quantity "equilibrium charge" introduced in lines 78 needs to be better defined there. Are you talking about charge or space charge density or space charge density per unit mass of dust? It is made clear later in the paper what you are measuring but this needs to be clarified in the Introduction, given the importance of the equilibrium concept throughout the work. Regarding hypotheses for equilibrium charge, the Introduction gives nothing and lines 195-196 gives nothing. Only late in the paper (Section 4.2) is any discussion provided. If this came in the Introduction, the reader would have a better idea where you were heading in the overall work.

Regarding one working hypothesis: dielectric breakdown, there is an important observational test: Corona discharge is a form of dielectric breakdown and furthermore, this process is a source of light. With a sensitive video camera operating in nighttime conditions (with better signal-to-noise ratio), one could look for light intensification as a signature for equilibrium. Have the authors tried this?

(3) Physical units

The authors should be clear about physical units for rho, M10, mu, lambda and ACD, all linked with equation (1) and (2) (where rho has standard MKS units of C/m^3.) It should also be made clear what ACD actually stands for. This may be Chinese, but

in any case needs to be spelled out because in my experience this is non-standard usage.

(4) Sign convention on Ez and polarity of space charge

Important missing information in this study is the sign convention on Ez and the predominant polarity of the space charge density. Figure 2 can't be interpreted without this information. (See again Williams et al., 2009)). This issue is also related to physical mechanisms for macroscopic dust particle charging and two prominent ones are as follows:

(i) Collisions between large and small particles in the cloud with selective charge transfer and then separation of the large and small (oppositely charged) particles by gravity. Result: a bipolar dust cloud

(ii) Lofting of fine dust particles by wind-driven saltation. Result: a unipolar cloud.

What can the authors offer up to distinguish these two mechanisms?

It is worth commenting further on findings by the reviewer that went beyond the published findings in Williams et al. (2009) and which are also based on work in Niger. This evidence came from a single day characterized by very gusty straight line winds, but of insufficient strength and persistence to form a deep opaque dust cloud. But with every strong gust, large perturbations (many kV/m and as a large as during the large haboob events) in the surface electric field were noted. This we take as evidence for mechanism (ii) above. The very find dust (clay) is charged with negative polarity during saltation. But in the context of the present work with emphasis on mass loading, please note Figure 6 in Williams et al. (2009) that does show some (weak) positive correlation between maximum E field and (inferred) mass loading. More analysis of this kind is needed in the present work to shed further light on physical mechanisms of dust charging.

(5) Puzzlements about Table 1

Table 1 is a reliable compilation of numbers for the ten documented cases, but would benefit from ACD values and maximum Ez values. But in light of claims that larger RH increased the charge transfer (contrary to this reviewer's intuition and experience in Niger where slightly more moisture and humidity served to suppress the dust and particularly the fine dust). I looked at extreme cases in Table 1. Case #2 has the largest RH and the largest rho, and Case #9 has the lowest RH and the largest rho. These findings are in keeping with my intuition. But then in studying in more detail the multi-regression and the evidence in Figures 4 and 5 I became confused. Sometimes the signs of the derivatives are positive and sometimes negative. The work should strive to go beyond regression to address physical explanations for behavior, whenever that is possible. And regarding regression alone, unless the coefficients are provided in equation (5), the reader does not have a quantitative result.

(6) Evidence for equilibrium effects

The equilibrium charge is a key concept in the paper. But when all is said and done, what exactly are the authors pointing to in support of such an effect? For example, in Figure 4, the space charge density is increasing monotonically with mass loading throughout the range, with no evidence for saturation. There are also no signs of asymptoting in Figure 5. What then is the real evidence for equilibrium charge?

Detailed edits/comments on the text:

Page 2 Line 24 Why is 10 min an important time scale?

Line 28 A little confusing as you never measure the charge on one dust particle in the paper.

Line 40 "electrical charge" Lines 41-42 This is not shown nor discussed later in the paper. Please explain why it is important? (It could be another explanation for the equilibrium charge, for example.)

[Figure]

Line 50 "of the electric field"

Line 64 "influence"

Line 71 "using a Faraday cage" It is not clear how you are measuring this quantity with a Faraday cage.

Line 80 "in the quantification of particle electrification"; "such an electrification equilibrium exists under…"

Line 84 "such as the ambient"

Line 86 change "such as" to "and especially"

Line 90 The authors do it with multi-regression but do not do it physically.

Line 110 What is a prevailing wind route?

Line 113 Why is this? I don't follow the argument.

Line 117 , 118 Vertical gradients in what quantity?

Page 6 Line 123 could add "at centrally-located S9"

Line 125 "by a solar panel system"

Figure 1 should make it clear that Ex and Ey are non-zero because you are measuring them in altitude above the surface

Line 140 "can be determined"

Line 143 It is not clear how you do your calibrations with instruments at this height

Line 158 You should give the sampling frequency.

Line 162 "The PM10 mass concentration..."

Line 166 "a sand particle"

Line 167 "a temperature-humidity sensor"

Line 170 Tell the scale over which the visibility measurement is made

Line 171 Presumably the Ez measurements are more frequent than 1 Hz.

Line 195 How did the SLR model show equilibrium effects?

Line 199 See Williams et al. (2009)

Line 213-214 Authors should make it clear that the derivatives will be shown to be both positive and negative.

Line 228 This is a HUGE field to have near the ground, and I would expect lots of corona light from ground features.

Figure 2 Reader needs the convention for Ez polarity to get the polarity of the dust cloud. Line 11

Please add the suggested quantities to Table 1. Visibility numbers are also shown in Williams et al. (2009)

Line 157 It is difficult to see the arrow directions on these plots. Page 13

Line 266 It is not clear to the reviewer that a constant ACD value is evidence for equilibrium charge unless that constant shows up in all cases. Has this been shown? And where has it been shown that ACD is independent of wind speed?

[Figure]

This figure 4 shows evidence that rho is increasing with RH. This runs contrary to my intuition. Page 15

Lines 312-313 This is not the scale that I got in looking at the figure. Those scales are larger.

Line 323 What lab studies show this?

Lines 347-349 What is evidence for this in the paper?

Line 366 Where do I see this finding in plots in the paper?

Lines 423-424 Where is this shown in the paper?

References

Suggest adding Williams et al. (2009) and studying it.

End review
* * *

---

## Author Comment (AC1) · 17 Oct 2018

**Responses to Jonathan Merrison's comments (RC1):**

Thanks are extended to the editor, Paola Formenti, and to the reviewers, Jonathan Merrison and an anonymous reviewer, for their careful work and thoughtful suggestions that greatly improved the manuscript.

The following text contains the reviewer's comments (black), our replies (blue) and the changes made to the manuscript (red).

**Comment 01:** This paper presents the results and analysis of a field campaign studying the electric field generated by a series of dust storms. This work is of high interest to those studying Aeolian processes, atmospheric electricity, aspects of planetology and especially the electrification of aerosols and granular materials. This is one of only a few such studies and it appears to be thorough and in depth applying a range of experimental sensor techniques.

**Response:**

We thank the reviewer for this positive assessment of our manuscript.

**Comment 02:** Generally in this paper the distinction between sand and dust is unclear, an example is line 155 in which the size distribution of saltating particles (by definition these must be sand) are used to make arguments about dust events. By definition sand grains cannot be suspended and are transported at low altitude (typically <1m), cohesion effects for sand are typically negligable. Dust cannot saltate and it is unclear whether they can/do in fact collide while in suspension, it may be that if in contact dust grains will cohere (aggregate). It is my impression that this study focusses on atmospheric dust in which case this should be made clear.

**Response:**

As you pointed out that our study is mainly concerned with airborne dust particles, but in the original manuscript, the size distributions of saltating particles rather than airborne dust particles are used to describe the observed dust storms. In the revised manuscript, we have added the measured size distributions of airborne

dust particles collected at the S9 site (5 m above the ground), and we can see that dust events occurring in the QLOA site have a very similar dust particle size distribution. The related sentence in the revised manuscript has been modified as:

Measurements of the size distribution of airborne dust particles (Fig. S2) and saltating particles (Fig. S3) implies that the dust events occurring in the QLOA site have a very similar particle size distribution.

In addition, the size distributions of airborne dust particles have been provided in Fig. S2 in the Supplement, as follows:

[Figure]

**Fig. S2.** Size distributions of the airborne dust particles collected at the S9 site (5 m above the ground). (a) A dust collector was mounted on a horizontally orientated steel bar. (b) Number distribution of the collected airborne dust particles during No. 01 and No. 02-10 dust storms. (c) The corresponding volume distribution of the collected airborne dust particles. Particle size analysis was performed using the Microtrac S3500 tri-laser particle size analyzer. Since the collected airborne dust particles of single dust storms are very few (i.e. No. 02-10 events), it is difficult to measure the size distribution of single dust storms by the collected dust sample. Consequently, the collected dust particles from No. 02-10 dust storms were combined to obtain a mean size distribution, as shown in Figs. S2a and S2b.

**Comments 03:** It is of some interest whether the field measurements made here are consistent with dust being typically electrified negatively, the authors might want to comment upon this.

**Response:**

In our field campaign, the observed space charge density at 5 m height is positive, which is consistent with the previous studies such as Kamra (1972) and Williams et al. (2009). Actually, the charge structure of dust storms is generally multipolar. In the revised manuscript, we have discussed this topic in detail as follows:

Previous measurements have demonstrated that the charge structure of dust clouds in dust storms could appear as unipolar, bipolar, and even multipolar. For example, Williams et al. (2009) measured the vertical E-field in dust storms and found both upward- and downward-pointing vertical E-field. They inferred that the dust cloud is unipolar if the near-ground particle charge transfer is dominating, while the dust cloud is bipolar if upper-air (volume) charge transfer is dominating. Direct dust storm charge measurements by Kamra (1972) have also observed both positive and negative space charge at 1.25 m height above the ground. Additionally, our recent dust storm E-field measurements up to a height of 30 m have shown that dust cloud could be multipolar (Zhang et al., 2017). In this study, the derived space charge density at 5 m height is positive, which is certainly reasonable, although many studies have observed a negative space charge. In fact, the charge structure of dust storms is closely associated with the transport of dust particles. There is no doubt that the large-scale and very-large-scale motions of flow exist in the high Reynolds number atmospheric surface layer (Hutchins et al., 2012), affecting the transport of dust particles because of dust following wind flow exactly (Jacob and Anderson, 2016). We can expect that a bipolar charge structure in each large-scale motions is produced by the bi-disperse suspensions of oppositely charged particles (Renzo and Urzay, 2018). Consequently, the multipolar charge structure of dust storms is formed by a series of bipolar charge of large-scale motions.

References:

Hutchins, N., Chauhan, K., Marusic, I., Monty, J., and Klewicki, J.: Towards reconciling the large-scale structure of turbulent boundary layers in the atmosphere and laboratory, Boundary-Layer Meteorol., 145, 273-306, https://doi.org/10.1007/s10546-012-9735-4, 2012.

Jacob, C., and Anderson, W.: Conditionally averaged large-scale motions in the neutral atmospheric boundary layer: Insights for aeolian processes, Boundary-Layer Meteorol., 162, 21-41, https://doi.org/10.1007/s10546-016-0183-4, 2016.

Renzo, M. D., and Urzay, J.: Aerodynamic generation of electric fields in turbulence laden with charged inertial particles, Nat. Commun., 9, 1676, https://doi.org/10.1038/s41467-018-03958-7, 2018.

**Comments 04:** The consecutive acquisition of charge (Line 73-80) through collisions and a so called equilibrium charge has been observed only in some laboratory studies (mostly using sand sized particles), it is not demonstrated in all experiments and is not generally accepted that electrification of dust in fact involves multiple collisions. Recent laboratory work implies that it is not (e.g. Alois et al. 2017, 2018). Similarly it seems that the work presented here is not in fact in agreement with a model based upon multiple collisions (e.g. line 259, 406). (Alois, S., Merrison, J., Iversen, J.J., Sesterhenn, J., (2017) Contact electrification in aerosolized monodispersed silica microspheres quantified using laser based velocimetry, Journal of Aerosol Science 106 1–10., Alois, S., Merrison, J., Iversen, J.J., Sesterhenn, J., (2018), Quantifying the contact electrification of aerosolized insulating particles, Powder Technology 332, 106–113)

**Response:**

We are sorry for our negligence of recent important studies associated with contact electrification of micron-sized particles (e.g. Alois et al. 2017, 2018). Alois et al. (2017) simultaneously measured the size and electrical charge of individual micron particles using a novel technique based on laser velocimetry and found that the surface charge density of the charged particles closely distributed around a constant value of 0.02 mC m$^{-2}$, which was probably caused by the electron field emission at particle contact site. According to the reviewer's suggestion, the changes made in the revised manuscript are threefold:

(i) line 73-83 has been modified as follows:

Furthermore, a large number of theoretical work and laboratory experiments have shown that the net electrical charge on millimeter-sized particles increases with increasing number of collisions/contacts and correlates with the particles' kinetic energy (Apodaca et al., 2010; Harper and Dufek, 2016; Matsuyama and Yamamoto, 1995; Zhang et al., 2013) until particles acquired a certain amount of charge (termed as equilibrium charge). The magnitude of the eventual equilibrium charge on particles is found to be independent of the particles' collisional dynamics and therefore will reduce the difficulties in the quantification of particle electrification. However, whether such an electrification equilibrium exists under natural circumstances, especially for micron-sized dust particles in dust storms, is unclear and needs to be verified.

(ii) The references of Alois et al. (2017, 2018) have been cited in the revised manuscript. That is, in section 4.2 we have added the following text:

More recently, a study of the contact electrification of micron-sized silica particles have also demonstrated that the surface charge density of the charged particles closely distributed around a constant value of 0.02 mC m-2 (that is, particles having a saturation or equilibrium charge), which is most possibly caused by the alternative mechanisms such as electron field emission and charge spreading at the particle contact site (Alois et al., 2017).

(iii) Line 342-344 in the manuscript has been modified as follows:

The proposed reasons for this are twofold: On one hand, the presence of adsorbed water could increase surface conductivity and particle-particle effective contact area, thus facilitating the ion or electron transfer (McCarty and Whitesides, 2008; Alois et al., 2018); On the other hand, $OH^-$ ions in adsorbed surface water could also act as charge carrier (Gu et al., 2013; Lacks and Sankaran, 2011; McCarty and Whitesides, 2008).

References:
Alois, S., Merrison, J., Iversen, J. J., and Sesterhenn, J.: Contact electrification in

aerosolized monodispersed silica microspheres quantified using laser based velocimetry, J. Aerosol Sci., 106, 1-10, https://doi.org/10.1016/j.jaerosci.2016.12.003, 2017.

Alois, S., Merrison, J., Iversen, J. J., and Sesterhenn, J.: Quantifying the contact electrification of aerosolized insulating particles, Powder Technol., 332, 106-113, https://doi.org/10.1016/j.powtec.2018.03.059, 2018.

**Comments 05:** As the authors point out in laboratory studies it appears that the charge concentration (per surface area) is a more useful physical parameter than charge to mass ratio ($\mu$), e.g. line is it possible to derive such a parameter from these measurements?. Alternatively information of charge per dust particle might in this case also be valuable.

**Response:**

To the best of our knowledge, both surface charge density and charge-to-mass ratio are very important physical quantities for the electrification of granular materials. As the previous studies found that surface charge density played a key role in investigating particle charging mechanism (Alois et al., 2017; Lacks and Sankaran, 2011; Merrison, 2012). And charge-to-mass ratio is very critical for determining the effects of electrostatic forces on particle motion (transport), because the acceleration of dust particle due to electrostatic forces can be described as $c\mathbf{E}$ (where $c$ is the charge-to-mass ratio and $\mathbf{E}$ is the electric field).

In this study, we cannot obtain the surface charge density and charge per particle because we only measured the mean space charge density and mass density (the information of the number of dust particles and real-time size distribution cannot be obtained).

**Comments 06:** It is of great interest that the observations presented here show a dependence upon RH especially as stated by the authors that the composition of the dust (soil) might imply a sensitivity to surface moisture (line 309). As the authors also point out some studies demonstrate dependence upon RH and others do not (line 337-347). Recent work has also shown that electrification can occur at extremely low RH

but that high RH may greatly enhance electrification for some materials. This appears to present a consistent picture (Alois 2018).

**Response:**

According to the reviewer's comments, we have cited the related recent work (i.e. Alois et al., 2018) in section 4.2 in the revised manuscript, as follows:

While water is not necessary for contact electrification (Baytekin et al., 2011a), a variety of studies indicated that such charge separation was strongly dependent on the RH (Esposito et al., 2016; McCarty and Whitesides, 2008; Xie and Han, 2012; Alois et al., 2018; Zhang et al., 2017). The proposed reasons for this are twofold: On one hand, the presence of adsorbed water could increase surface conductivity and particle-particle effective contact area, thus facilitating the ion or electron transfer (McCarty and Whitesides, 2008; Alois et al., 2018); On the other hand, OH− ions in adsorbed surface water could also act as charge carrier (Gu et al., 2013; Lacks and Sankaran, 2011; McCarty and Whitesides, 2008).

---

## Author Comment (AC2) · 17 Oct 2018

**Responses to Anonymous Reviewer #2's comments (RC2):**

Thanks are extended to the editor, Paola Formenti, and to the reviewers, Jonathan Merrison and an anonymous reviewer, for their careful work and thoughtful suggestions that greatly improved the manuscript.

The following text contains the reviewer's comments (black), our replies (blue) and the changes made to the manuscript (red).

**Comment 01:** This is a well-organized study of natural dust storm electrification, with novel analysis and new findings. The English has been meticulously prepared. Some improvements are in order pertaining to the physical interpretation and the real evidence for equilibrium effects. A number of substantive issues are worth addressing by the authors in the preparation of their final manuscript. These issues are followed by detailed edits/comments on the text.

Summary: Publish after appropriate revision

Response:

We thank the reviewer for this positive assessment of our manuscript.

**Comments 02:** Physical origin of dust events. The physical/meteorological basis for the events with other extensive documentation in this work is not elaborated. Lines 110-111 suggest a role for straight line winds. Are the cold downdrafts from thunderstorms/squall lines important for these events, as was the case in Niger in a study by Williams et al. (Atmos. Res., 2009). (We are aware of earlier thunderstorm studies in the Lanzhou area of China by other investigators-S. Liu for example.)

**Response:**

Indeed, dust storms can be caused by various weather systems, such as the monsoon winds, cyclones, and thunderstorms/squall lines, depending on where dust storms occur (Shao, 2000; Williams et al., 2009). In this study, due to the lack of the meteorological data, we cannot determine the physical origin of the observed dust events. However, previous studies have shown that from March to May, the Gobi

region (including QLOA site) is mainly affected by Mongolian cyclones (please see Chapter 2 in Shao, Y.: *Physics and modelling of wind erosion*, Springer-Verlag, Netherlands, 2000.). Consequently, According to the reviewer's comments, we have added a description on the topic of "Physical origin of dust events", as follows:

The area was selected since it lies within a dusty belt in the Hexi Corridor (Wang et al., 2018), which is mainly affected by the Mongolian cyclones (and probably by the cold downdrafts from thunderstorms/squall) during the observational period and is therefore frequently subjected to dust events (Shao, 2000; Williams et al., 2009).

**Comments 03:** Physical hypotheses for "equilibrium effects". First of all, the physical quantity "equilibrium charge" introduced in lines 78 needs to be better defined there. Are you talking about charge or space charge density or space charge density per unit mass of dust? It is made clear later in the paper what you are measuring but this needs to be clarified in the Introduction, given the importance of the equilibrium concept throughout the work. Regarding hypotheses for equilibrium charge, the Introduction gives nothing and lines 195-196 gives nothing. Only late in the paper (Section 4.2) is any discussion provided. If this came in the Introduction, the reader would have a better idea where you were heading in the overall work.

Regarding one working hypothesis: dielectric breakdown, there is an important observational test: Corona discharge is a form of dielectric breakdown and furthermore, this process is a source of light. With a sensitive video camera operating in nighttime conditions (with better signal-to-noise ratio), one could look for light intensification as a signature for equilibrium. Have the authors tried this?

**Response:**

For this comment, the responses include two aspects:

(i) We are very sorry for our negligence of the clear definition of "equilibrium charge" in the Introduction. According to the reviewer's suggestion, we have added the following descriptions in the Introduction:

The ratio of space charge density to the dust mass concentration (called mean

charge-to-mass ratio of dust particles) rather than charge on the individual particles is generally used to quantify the electrical properties of dusty phenomena. In this study, we build on a set of field observations through an extensive statistical analysis to assess the mean scaled charge-to-mass ratio of airborne dust particles $\mu^*$ (defined in Sect. 2.2) in dust storms and to untangle the influences of environmental factors (i.e., ambient temperature and RH) on the $\mu^*$. Therefore, an electrification equilibrium is said to be built if $\mu^*$ remains constant at the given ambient temperature and RH.

(ii) As the reviewer pointed out that corona discharge could form at highly curved regions on instruments, such as sharp corners, projecting points, edges of metal surfaces, or small diameter wires (because the E-fields is up to ~100 kV/m during dust storms). Unfortunately, we did not observe the "corona discharge" effects in nighttime conditions. It is worthwhile to perform such observations in the future works.

**Comments 04:** Physical units. The authors should be clear about physical units for rho, M10, mu, lambda and ACD, all linked with equation (1) and (2) (where rho has standard MKS units of C/m^3.) It should also be made clear what ACD actually stands for. This may be Chinese, but in any case needs to be spelled out because in my experience this is non-standard usage.

**Response:**

Thanks for the reviewer's important suggestions. In the revised manuscript, we have re-defined the ACD as a new physical quantity "scaled mean charge-to-mass ratio", which is physically meaningful and clear. Meanwhile, the physical units of all quantities have been unified in order to ensure that all equations in this study are dimensionally homogeneous. The following changes have been made in the revised manuscript, that is:

Consequently, the scaled mean charge-to-mass ratio $\mu^*$, which is a common measure of the charge-to-mass ratio of dust particles, can be defined as

$$\mu^* \equiv \frac{\rho}{M_{10}} \tag{1}$$

where $\mu^*= \mu/\lambda$ (λ is assumed to be a constant) is equal to the mean charge-to-mass ratio divided by the $PM_{10}$ mass fraction. From this definition, $\mu^*$ can be determined once the space charge density and $PM_{10}$ mass concentration have been determined. By adopting Standard International units, the units of $\mu^*$ and $\mu$ are C $kg^{-1}$, the unit of ρ is C $m^{-3}$, the unit of $PM_{10}$ concentration $M_{10}$ is kg $m^{-3}$, and the $PM_{10}$ mass fraction λ has a dimensional unit in Eq. (1).

**Comments 05**: Sign convention on Ez and polarity of space charge. Important missing information in this study is the sign convention on Ez and the predominant polarity of the space charge density. Figure 2 can't be interpreted without this information. (See again Williams et al., 2009)). This issue is also related to physical mechanisms for macroscopic dust particle charging and two prominent ones are as follows:

(i) Collisions between large and small particles in the cloud with selective charge transfer and then separation of the large and small (oppositely charged) particles by gravity. Result: a bipolar dust cloud.

(ii) Lofting of fine dust particles by wind-driven saltation. Result: a unipolar cloud.

What can the authors offer up to distinguish these two mechanisms?

It is worth commenting further on findings by the reviewer that went beyond the published findings in Williams et al. (2009) and which are also based on work in Niger. This evidence came from a single day characterized by very gusty straight line winds, but of insufficient strength and persistence to form a deep opaque dust cloud. But with every strong gust, large perturbations (many kV/m and as a large as during the large haboob events) in the surface electric field were noted. This we take as evidence for mechanism (ii) above. The very find dust (clay) is charged with negative polarity during saltation. But in the context of the present work with emphasis on mass loading, please note Figure 6 in Williams et al. (2009) that does show some (weak) positive correlation between maximum E field and (inferred) mass loading. More analysis of this kind is needed in the present work to shed further light on physical mechanisms

of dust charging.

**Response:**

Thanks for the reviewer's very important suggestions. According to the reviewer's suggestions, the descriptions added in the revised manuscript include:

(i) the "sign convention of E-fields" has been added in the caption of Figure 2:

The E-fields $E_x$, $E_y$, and $E_z$ are positive if they point in the positive directions of $x$, y, and z axes depicted in Fig. 1. That is, $E_z$ and fair-weather atmospheric E-field are oppositely directed.

(ii) the discussions of "the polarity of the space charge density and physical mechanisms" of dust charging have been added in section 4.1 of the revised manuscript as follows:

Previous measurements have demonstrated that the charge structure of dust clouds in dust storms could appear as unipolar, bipolar, and even multipolar. For example, Williams et al. (2009) measured the vertical E-field in dust storms and found both upward- and downward-pointing vertical E-field. They inferred that the dust cloud is unipolar if the near-ground particle charge transfer is dominating, while the dust cloud is bipolar if upper-air (volume) charge transfer is dominating. Direct dust storm charge measurements by Kamra (1972) have also observed both positive and negative space charge at 1.25 m height above the ground. Additionally, our recent dust storm E-field measurements up to a height of 30 m have shown that dust cloud could be multipolar (Zhang et al., 2017). In this study, the derived space charge density at 5 m height is positive, which is certainly reasonable, although many studies have observed a negative space charge. In fact, the charge structure of dust storms is closely associated with the transport of dust particles. There is no doubt that the large-scale and very-large-scale motions of flow exist in the high Reynolds number atmospheric surface layer (Hutchins et al., 2012), affecting the transport of dust particles because of dust following wind flow exactly (Jacob and Anderson, 2016). We can expect that a bipolar charge structure in each large-scale motions is produced by the bi-disperse suspensions of oppositely charged particles (Renzo and Urzay, 2018). Consequently,

the multipolar charge structure of dust storms is formed by a series of bipolar charge of large-scale motions.

References:

Hutchins, N., Chauhan, K., Marusic, I., Monty, J., and Klewicki, J.: Towards reconciling the large-scale structure of turbulent boundary layers in the atmosphere and laboratory, Boundary-Layer Meteorol., 145, 273-306, https://doi.org/10.1007/s10546-012-9735-4, 2012.

Jacob, C., and Anderson, W.: Conditionally averaged large-scale motions in the neutral atmospheric boundary layer: Insights for aeolian processes, Boundary-Layer Meteorol., 162, 21-41, https://doi.org/10.1007/s10546-016-0183-4, 2016.

Renzo, M. D., and Urzay, J.: Aerodynamic generation of electric fields in turbulence laden with charged inertial particles, Nat. Commun., 9, 1676, https://doi.org/10.1038/s41467-018-03958-7, 2018.

**Comments 06**: Puzzlements about Table 1. Table 1 is a reliable compilation of numbers for the ten documented cases, but would benefit from ACD values and maximum Ez values. But in light of claims that larger RH increased the charge transfer (contrary to this reviewer's intuition and experience in Niger where slightly more moisture and humidity served to suppress the dust and particularly the fine dust). I looked at extreme cases in Table 1. Case #2 has the largest RH and the largest rho, and Case #9 has the lowest RH and the largest rho. These findings are in keeping with my intuition. But then in studying in more detail the multi-regression and the evidence in Figures 4 and 5 I became confused. Sometimes the signs of the derivatives are positive and sometimes negative. The work should strive to go beyond regression to address physical explanations for behavior, whenever that is possible. And regarding regression alone, unless the coefficients are provided in equation (5), the reader does not have a quantitative result.

**Response:**

For this comment, the changes made in the revised manuscript are threefold:

(i) Changes associated with Table 1:

According to the reviewer's suggestion, the maximum values of the scaled charge-to-mass ratio (i.e. ACD in the original manuscript) and $E_z$ have been added in

Table 1 as follows:

[revised manuscript text omitted]

(iii) Quantitative result for "temperature and RH dependence of $\mu^*$":

To show the quantitative result clearly, we have added the quantitative relationships between $\mu^*$, $T_a$, and RH in section 3.2. That is:

$$\mu^* = (26955 - 2719T_a - 698T_aRH + 89T_a^2 + 60950RH^2 + 24T_aRH) \times 10^{-4} \quad (6)$$

**Comments 07**: Evidence for equilibrium effects. The equilibrium charge is a key concept in the paper. But when all is said and done, what exactly are the authors pointing to in support of such an effect? For example, in Figure 4, the space charge density is increasing monotonically with mass loading throughout the range, with no evidence for saturation. There are also no signs of asymptoting in Figure 5. What then is the real evidence for equilibrium charge?

**Response:**

In this study, we have defined a physical quantity, scaled mean charge-to-mass ratio $\mu^* \equiv \rho / M_{10}$, to assess the electrical properties of dust storms. An electrification equilibrium is said to be built if $\mu^*$ remains constant (in other words, $\rho$ and $M_{10}$ are linearly correlated) at the given ambient temperature and RH. The linear relationship is quantified by the squared wavelet coherence $R^2(n, s)$ in time and frequency space, which can be thought of as a localized correlation coefficient between two time series in time and frequency space. In this study, by performing the wavelet coherence analysis, we found that $\rho$ and $M_{10}$ were significantly correlated over the 10 min timescales. Meanwhile, in Fig. 4, the plots of the 10 min moving average of $\rho$ vs. $M_{10}$ at given ambient temperature and RH have shown that the slopes (i.e. $\mu^*$) are nearly constant. These are the evidence of large-scale "electrification equilibrium". Note that once ambient temperature or RH is changed, the large-scale system will reach a new electrification equilibrium. Thus, we used a multiple linear regression model to quantify the temperature and RH dependence of $\mu^*$, and the results are shown in Fig. 5.

This issue has been discussed in detail in the revised manuscript. For example, in section 3.1 "Electrification equilibrium effects over large timescales", the related sentences are:

To quantify the strong large timescale correlations between $\rho$ and $M_{10}$, we performed SLR analysis between the 10 min moving average (See Fig. S6 in the Supplement) of the $\rho$ and $M_{10}$ time series, where the fitted linear regression slope is equal to the $\mu^*$. The SLR analysis was performed for a set of given temperature and

RH intervals (within 2 °C and 2 %). As shown in Fig. 4, there is a significant linear relationship between ρ and $M_{10}$ at a given ambient temperature and RH (with median $R^2$ of ~0.71-0.98 and p<0.01, see Fig. S7 and Table S1 in the Supplement), suggesting that $\mu^*$ is nearly constant during a period that ambient temperature and RH are fixed. The long period constant $\mu^*$ implies that electrification equilibrium has been established (on average) where the rates of gain and loss of electrical charge are equal. $\mu^*$ is significantly influenced by environmental factors but independent of the particles' collisional dynamics and wind speed.

In section 4.2 "The physical mechanisms for electrification equilibrium", the related sentences are given in the revised manuscript:

In the present study, the large-scale electrification equilibrium effects widely exist in dust storms (Figs. 3 and 4). However, in dust storms, we propose that such electrification equilibrium of a large-scale system (averaged over multi-cubic meter volume and 10 min) is a dynamic equilibrium rather than the saturation of individual particles. In this case, the charges on dust particles transfer between the large-scale systems at an equal rate, meaning there is no net charge exchange. Charge transfer between individual dust particles may in fact occur, but to such an extent that we cannot observe the changes in $\mu^*$ of the large-scale system under certain ambient condition. It should be emphasized that the concept of large-scale electrification equilibrium is only applied to the dust storms under certain ambient condition; that is, $\mu^*$ is constant with varying particles' dynamics at given temperature and RH. Once ambient temperature or RH is changed, the large-scale system will reach a new electrification equilibrium. Consequently, such equilibrium can be termed environmental-dependent equilibrium effects.

**Comment 08:** Page 2: Line 24 Why is 10 min an important time scale?

**Response:**

The integral time scale T of turbulence is an important concept for aeolian transport, which is around ~10 min in the atmospheric boundary layer and ~1 s in the

wind-tunnel (please see Durán et al., 2011 for the details). The wind variations over time scales smaller than T are attributed to turbulence, while variations over time scale larger than T are attributed to meteorological effects. In general, the aeolian transport and wind strength are highly correlated over time scales larger than T. Since the fine dust particles often follow the wind strictly, thus $\rho$ and $M_{10}$ are strongly correlated when both of them are averaged over T (the effects of turbulent fluctuation is excluded). We have added the description of the importance of the 10 min time scales in the revised manuscript. That is:

Actually, the integral time scale of atmospheric turbulence is on the order of ~10 min (Durán et al., 2011). The wind variations over time scales smaller than ~10 min are attributed to turbulence, while variations over time scales larger than ~10 min are attributed to meteorological effects. In general, the aeolian transport and wind strength are highly correlated over time scales larger than ~10 min. Since the fine dust particles often follow the wind strictly, the large timescale strong correlation between $\rho$ and $M_{10}$ are certainly reasonable where the effects of turbulent fluctuations are excluded.

**Comment 09:** Line 28 Alittle confusing as you never measure the charge on one dust particle in the paper.

**Response:**

To avoid this confusion, the statement of "…suggesting that the mean charge on dust particles…" has been revised as "…suggesting that the estimated mean charge on dust particles…"

**Comment 10:** Page 3: Line 40 "electrical charge"

**Response:**

The statement of "electrical charges" has been revised as "electrical charge"

**Comment 11:** Lines 41-42: This is not shown nor discussed later in the paper. Please

explain why it is important? (It could be another explanation for the equilibrium charge, for example.)

**Response:**

The statement of "The strong electrostatic forces exerted on dust particles, which are comparable to gravitational force, could considerably affect the motion of particles and facilitate the lifting of particles from the ground (Esposito et al., 2016; Harper et al., 2017; Kok and Renno, 2008; Schmidt et al., 1998; Zheng et al., 2003)." presented here is used to emphasize the importance of electrostatic forces, which is not related to the equilibrium charge. We prefer to retain such statement to better organize our "Introduction" of our manuscript.

**Comment 12:** Line 50 "of the electric field", Line 64 "influence"

**Response:**

The statements of "…of electric field…" and "…influences…" have been revised as "…of the electric field…" and "…influence…", respectively.

**Comment 13:** Page 4, Line 71: "using a Faraday cage" It is not clear how you are measuring this quantity with a Faraday cage.

**Response:**

The statement of "…using Faraday cage…" has been revised as "…using a Faraday cage…"

In this study, the charge-to-mass ratio was not measured by the Faraday cage. We quantify the electrical properties of airborne dust particles by the scaled charge-to-mass ratio $\mu^*$ (equivalent to the ACD defined in the original manuscript, and defined by Eq. 1 in the revised manuscript), which is determined indirectly by measuring the divergence of the electric field and dust concentration simultaneously. This related text in the "Introduction" has just summarized the existing measurements of charged saltating particles (the detailed measurement method can be found in the references of Bo et al., 2014 and Schmidt et al., 1998).

**Comment 14:** Line 80 "in the quantification of particle electrification"; "such an electrification equilibrium exists under…"

**Response:**

The statements of "…in particle electrification quantifications." and "…such electrification equilibrium effects exist under…" have been revised as "…in the quantification of particle electrification…" and "…such an electrification equilibrium exists under…",respectively.

**Comment 15:** Line 84 "such as the ambient", Line 86 change "such as" to "and especially"

**Response:**

The statements of "…such as ambient…" and "…such as…" have been revised as "…such as the ambient…" and "…and especially…", respectively.

**Comment 16:** Line 90 The authors do it with multi-regression but do not do it physically.

**Response:**

According to the reviewer's comments, we have added the physical explanations for "temperature and RH dependence of $\mu^*$" in section 4.2 as follows:

In addition, the equilibrium value ($\mu^*$) of the large-scale system was found to be strongly influenced by RH and ambient temperature in dust storms during our field observations. While water is not necessary for contact electrification (Baytekin et al., 2011a), a variety of studies indicated that such charge separation was strongly dependent on the RH (Esposito et al., 2016; McCarty and Whitesides, 2008; Xie and Han, 2012; Alois et al., 2018; Zhang et al., 2017). The proposed reasons for this are twofold: On one hand, the presence of adsorbed water could increase surface conductivity and particle-particle effective contact area, thus facilitating the ion or electron transfer (McCarty and Whitesides, 2008; Alois et al., 2018); On the other hand, $OH^-$ ions in adsorbed surface water could also act as charge carrier (Gu et al., 2013;

Lacks and Sankaran, 2011; McCarty and Whitesides, 2008). We also found that $\mu^*$ was strongly affected by the ambient temperature. This is consistent with other reports, which showed that the dielectric constant and conductivity of the adsorbed water were significantly linked to the ambient temperature (Gu et al., 2013; Lacks and Sankaran, 2011; Wei and Gu, 2015). As shown in Figs. 5b and 5c, the predicted $\mu^*$ is nonlinearly related to ambient temperature and RH. Specifically, the predicted $\mu^*$ increases at $T_a$=27.5 °C but decreases at $T_a$=5.5 °C with increasing RH. This result has also been verified by other studies (Xie and Han, 2012; Zheng et al., 2014). For example, by considering the effects of a water film on the particle-particle effective contact area, Zheng et al. (2014) revealed that the net charge transfer between two particles increased first then decreased with increasing RH. In addition, a wind-tunnel measurement found that the E-fields produced by charged sand particles increased first then decreased with increasing ambient temperature when RH=17 % (Xie and Han, 2012). Therefore, in Fig. 5b, the different patterns of $\mu^*$ at different ambient temperatures could be explained by the coupling effects between the nonlinear affecting factors ambient temperature and RH.

**Comment 17:** Page 5 Line 110: What is a prevailing wind route?

**Response:**

We are very sorry for this inappropriate statement. Considering the reviewer's comment, the statement of "…prevailing wind route…" has been revised as "…prevailing wind direction…"

**Comment 18:** Line 113: Why is this? I don't follow the argument.

**Response:**

As you pointed out that our study is mainly concerned with airborne dust particles, but in the original manuscript, the size distributions of saltating particles rather than airborne dust particles are used to describe the observed dust storms. In the revised manuscript, we have added the measured size distributions of airborne

dust particles collected at the S9 site (5 m above the ground), and we can see that dust events occurring in the QLOA site have a very similar particle size distribution. The related sentence in the revised manuscript has been modified as:

Measurements of the size distribution of airborne dust particles (Fig. S2) and saltating particles (Fig. S3) implies that the dust events occurring in the QLOA site have a very similar particle size distribution.

In addition, the size distributions of airborne dust particles are provided in Fig. S2 in the Supplement, as follows:

[Figure]

**Fig. S2.** Size distributions of the airborne dust particles collected at the S9 site (5 m above the ground). (a) A dust collector was mounted on a horizontally orientated steel bar. (b) Number distribution of the collected airborne dust particles during No. 01 and No. 02-10 dust storms. (c) The corresponding volume distribution of the collected airborne dust particles. Particle size analysis was performed using the Microtrac S3500 tri-laser particle size analyzer. Since the collected airborne dust particles of single dust storms are very few (i.e. No. 02-10 events), it is difficult to measure the size distribution of single dust storms by the collected dust sample. Consequently, the collected dust particles from No. 02-10 dust storms were combined to obtain a mean size distribution, as shown in Figs. S2a and S2b.

**Comment 19:** Line 117, 118 Vertical gradients in what quantity?

**Response:**

The gradients for E-fields have been measured in this study. Thus, the text has been modified as follows:

Among these towers, the main tower with a 32 m height could be used to measure the vertical E-field gradients, and the remaining 20 towers with 5 m height could be designed to determine the streamwise and spanwise gradients of E-fields (Fig. 1b).

**Comment 20:** Page 6 Line 123: could add "at centrally-located S9", Line 125: "by a solar panel system"

**Response:**

The statement "at centrally-located S9" has been added in the revised manuscript. "by the solar panel system" has been revised as "by a solar panel system"

**Comment 21:** Figure 1 should make it clear that Ex and Ey are non-zero because you are measuring them in altitude above the surface

**Response:**

The statement of "It is worth noting that the x and y components of E-fields are generally non-zero because dust transport is non-uniform in the horizontal plane (Zheng, 2013)." has been added in the caption of Figure 1 in the revised manuscript.

**Comment 22:** Line 140 "can be determined"

**Response:**

The statement of "can be estimated" has been revised as "can be determined"

**Comment 23:** Page 7 Line 143: It is not clear how you do your calibrations with instruments at this height.

**Response:**

The statements: "Before performing field measurements, all instruments were carefully calibrated in the laboratory. The VREFM sensors were also calibrated at QLOA site by comparing its output to a higher accuracy atmospheric E-field mill (see Fig. S7 in the Supplement). To achieve the best possible instrument accuracy, we performed re-calibration for VREFM sensors and periodic cleaning for Aerosol Monitor 8530EP twice a month during the observational period." were added in the revised manuscript for clarifying the instrument calibrations in our field observations.

**Comment 24:** Line 158: You should give the sampling frequency.

**Response:**

According to the reviewer's suggestion, sampling frequency has been given in Sect. 2.2, as follows:

All instruments were monitored continuously and simultaneously with a sampling frequency of 1 Hz (except for the CSAT3B which had a sampling frequency of 50 Hz)

**Comment 25:** Line 162: "The PM10 mass concentration…", Line 166: "a sand particle", Line 167: "a temperature-humidity sensor"

**Response:**

The statements of "$PM_{10}$ mass concentration", "sand particle", and "temperature-humidity sensor" have been revised as "The $PM_{10}$ mass concentration", "a sand particle", and "a temperature-humidity sensor", respectively.

**Comment 26:** Line 170 Tell the scale over which the visibility measurement is made.

**Response:**

According to the reviewer's suggestion, the statement of "…and visibility sensor (Model 6000, Belfort Instrument), measuring visibility ranging from 5 m to 10 km with $\pm$10 % accuracy…" has been added in the revised manuscript.

**Comment 27:** Line 171: Presumably the Ez measurements are more frequent than 1 Hz.

**Response:**

Indeed, Ez is measured with 1 Hz frequency. We have added the following description in the revised manuscript:

All instruments were monitored continuously and simultaneously with a sampling frequency of 1 Hz (except for the CSAT3B which had a sampling frequency of 50 Hz).

**Comment 28:** Line 195: How did the SLR model show equilibrium effects?

**Response:**

As we stated in the response of comment 07, we have defined a physical quantity, scaled mean charge-to-mass ratio $\mu^* \equiv \rho/M_{10}$, to assess the electrical properties of dust storms. An electrification equilibrium is said to be built if $\mu^*$ remains constant (in other words, $\rho$ and $M_{10}$ are linearly correlated) at the given ambient temperature and RH. The linear relationship is quantified by the squared wavelet coherence $R^2(n,s)$ in time and frequency space, which can be thought of as a localized correlation coefficient between two time series in time and frequency space. In this study, by performing the wavelet coherence analysis, we found that $\rho$ and $M_{10}$ were significantly correlated over the 10 min timescales. Meanwhile, in Fig. 4, the plots of the 10 min moving average of $\rho$ vs. $M_{10}$ at given ambient temperature and RH have shown that the slopes (i.e. $\mu^*$) are nearly constant. These are the evidence of large-scale "electrification equilibrium". Note that once ambient temperature or RH is changed, the large-scale system will reach a new electrification equilibrium. Thus, we used a multiple linear regression model to quantify the temperature and RH dependence of $\mu^*$, and the results are shown in Fig. 5. This issue has been discussed in detail in the revised manuscript (please see the response of comment 07 for the details).

**Comment 29:** Page 9 Line 199 See Williams et al. (2009)

**Response:**

The study of Williams et al. (2009) has been added in the revised manuscript.

**Comment 30:** Line 213-214: Authors should make it clear that the derivatives will be shown to be both positive and negative.

**Response:**

According to the reviewer's suggestion, we have added the statement of "The partial derivatives of E-fields were estimated from the interpolation-based numerical method and will be shown to be both positive and negative (see Fig. S5 in the Supplement)" in the revised manuscript.

**Comment 31:** Line 228: This is a HUGE field to have near the ground, and I would expect lots of corona light from ground features.

**Response:**

As the reviewer pointed out that corona discharge could form at highly curved regions on instruments, such as sharp corners, projecting points, edges of metal surfaces, or small diameter wires (because the E-fields is up to ~100 kV/m during dust storms). Unfortunately, we did not observe the "corona discharge" effects in nighttime conditions. It is worthwhile to perform such observations in the future works.

**Comment 32:** Figure 2: Reader needs the convention for Ez polarity to get the polarity of the dust cloud.

**Response:**

According to the reviewer's suggestion, we have added the convention of E-fields polarity in the caption of Fig. 2, as follows:

The E-fields $E_x$, $E_y$, and $E_z$ are positive if they point in the positive directions of $x$, $y$, and $z$ axes depicted in Fig. 1. That is, $E_z$ and fair-weather atmospheric E-field are oppositely directed.

Additionally, as we discussed in the response of comment 05, we have added the discussions of the polarity of the space charge density and physical mechanisms of

dust charging in section 4.1 of the revised manuscript as follows:

Previous measurements have demonstrated that the charge structure of dust clouds in dust storms could appear as unipolar, bipolar, and even multipolar. For example, Williams et al. (2009) measured the vertical E-field in dust storms and found both upward- and downward-pointing vertical E-field. They inferred that the dust cloud is unipolar if the near-ground particle charge transfer is dominating, while the dust cloud is bipolar if upper-air (volume) charge transfer is dominating. Direct dust storm charge measurements by Kamra (1972) have also observed both positive and negative space charge at 1.25 m height above the ground. Additionally, our recent dust storm E-field measurements up to a height of 30 m have shown that dust cloud could be multipolar (Zhang et al., 2017). In this study, the derived space charge density at 5 m height is positive, which is certainly reasonable, although many studies have observed a negative space charge. In fact, the charge structure of dust storms is closely associated with the transport of dust particles. There is no doubt that the large-scale and very-large-scale motions of flow exist in the high Reynolds number atmospheric surface layer (Hutchins et al., 2012), affecting the transport of dust particles because of dust following wind flow exactly (Jacob and Anderson, 2016). We can expect that a bipolar charge structure in each large-scale motions is produced by the bi-disperse suspensions of oppositely charged particles (Renzo and Urzay, 2018). Consequently, the multipolar charge structure of dust storms is formed by a series of bipolar charge of large-scale motions.

**Comment 33:** Line 11 Please add the suggested quantities to Table 1. Visibility numbers are also shown in Williams et al. (2009)

**Response:**

According to the reviewer's suggestion, we have added the maximum values of scaled charge-to-mass ratio $\mu^*_{max}$ and vertical E-field intensity $E_{z,max}$, in Table 1. Please see the response of comment 06 for the details.

**Comment 34:** Line 157: It is difficult to see the arrow directions on these plots.

**Response:**

To better show the wavelet coherence, we have removed the arrows in Fig. 3. In addition, the relative phase relationships (denotes by arrows) are currently shown in Figs. S9-S13. Please see Fig. 3 in the revised manuscript and Figs. S9-S13 in the Supplement for the details.

**Comment 35:** Page 13 Line 266: It is not clear to the reviewer that a constant ACD value is evidence for equilibrium charge unless that constant shows up in all cases. Has this been shown? And where has it been shown that ACD is independent of wind speed?

**Response:**

As we discussed in the response of comment 07, in this study, the concept of large-scale electrification equilibrium is only applied to the dust storms under certain ambient condition; that is, $\mu^*$ is constant with varying particles' dynamics at given temperature and RH. Once ambient temperature or RH is changed, the large-scale system will reach a new electrification equilibrium. Consequently, such equilibrium can be termed environmental-dependent equilibrium effects. For example, in Fig. (4a) $\rho$ and $M_{10}$ are linearly correlated (the ratio of $\rho$ to $M_{10}$, slope, is constant); however, in Fig. (4c) there is a new linear relationship between them. In addition, the linear relationship between $\rho$ and $M_{10}$ (e.g. Fig. 4c) is independent of the variation of wind speed (e.g. Fig. 4d). Please see the response of comment 07 for the details.

**Comment 36:** Page 14: This figure 4 shows evidence that rho is increasing with RH. This runs contrary to my intuition.

**Response:**

As we discussed in the response of comment 06, most previous studies found that charge transfer processes are nonlinearly related to ambient temperature and RH (Lacks and Sankaran, 2011; McCarty and Whitesides, 2008; Xie et al., 2012; Zheng et al., 2014). For example, by considering the effects of a water film on the particleparticle effective contact area, Zheng et al. (2014) revealed that the net charge transfer between two particles increased first then decreased with increasing RH. In addition, a wind-tunnel measurement found that the E-fields produced by charged sand particles increased first then decreased with increasing ambient temperature when RH=17 % (Xie and Han, 2012). We have added extensive discussions on this topic in the revised manuscript as follows:

In addition, the equilibrium value ($\mu^*$) of the large-scale system was found to be strongly influenced by RH and ambient temperature in dust storms during our field observations. While water is not necessary for contact electrification (Baytekin et al., 2011a), a variety of studies indicated that such charge separation was strongly dependent on the RH (Esposito et al., 2016; McCarty and Whitesides, 2008; Xie and Han, 2012; Alois et al., 2018; Zhang et al., 2017). The proposed reasons for this are twofold: On one hand, the presence of adsorbed water could increase surface conductivity and particle-particle effective contact area, thus facilitating the ion or electron transfer (McCarty and Whitesides, 2008; Alois et al., 2018); On the other hand, OH$^-$ ions in adsorbed surface water could also act as charge carrier (Gu et al., 2013; Lacks and Sankaran, 2011; McCarty and Whitesides, 2008). We also found that $\mu^*$ was strongly affected by the ambient temperature. This is consistent with other reports, which showed that the dielectric constant and conductivity of the adsorbed water were significantly linked to the ambient temperature (Gu et al., 2013; Lacks and Sankaran, 2011; Wei and Gu, 2015). As shown in Figs. 5b and 5c, the predicted $\mu^*$ is nonlinearly related to ambient temperature and RH. Specifically, the predicted $\mu^*$ increases at $T_a$=27.5 °C but decreases at $T_a$=5.5 °C with increasing RH. This result has also been verified by other studies (Xie and Han, 2012; Zheng et al., 2014). For example, by considering the effects of a water film on the particle-particle effective contact area, Zheng et al. (2014) revealed that the net charge transfer between two particles increased first then decreased with increasing RH. In addition, a wind-tunnel measurement found that the E-fields produced by charged sand particles increased first then decreased with increasing ambient temperature when RH=17 % (Xie and Han,

2012). Therefore, in Fig. 5b, the different patterns of $\mu^*$ at different ambient temperatures could be explained by the coupling effects between the nonlinear affecting factors ambient temperature and RH.

**Comment 37:** Page 15 Lines 312-313: This is not the scale that I got in looking at the figure. Those scales are larger.

**Response:**

We are very sorry for our inappropriate statement. According to the reviewer's suggestion, the statement has been revised as "The VREFMs spacing is respectively ~1.6, 5, and 10 m in the vertical, spanwise, and streamwise directions owing to the rapid variation of E-fields along the vertical direction and slow variation along the spanwise and streamwise directions (see Fig. S5 in the Supplement)"

**Comment 38:** Line 323: What lab studies show this?

**Response:**

A large number of studies showed that dust electrification might be attributed to electron transfer, ion transfer, material transfer mechanism. The saline-alkali soil at QLOA site may enhance ion transfer between dust particles (see McCarty and Whitesides, 2008 for the details). There are no direct laboratory experiments have demonstrated that the saline-alkali soil can enhance electrification of dust particles. Therefore, we have just changed the statement "which lead to high E-field intensity in dust storms" to "which may lead to high E-field intensity in dust storms" in this version of the revised manuscript. We plan to verify this issue in the future works.

**Comment 39:** Page 16 Lines 347-349: What is evidence for this in the paper?

**Response:**

The destruction of the large-scale equilibrium state is characterized by a weak correlation between space charge density and dust concentration (e.g., Figs. a-c in the following, as well as little R2 in Fig. S7).

[Figure]

(a)-(c): Coherence analyses between the space charge density and dust concentration. Dashed rectangular boxes denote the destructions of large-scale electrification equilibrium at some time.

**Comment 40:** Page 17 Line 366: Where do I see this finding in plots in the paper?

Page 19 Lines 423-424: Where is this shown in the paper?

**Response:**

For this comment, the responses include two aspects:

(i) From Fig. 5b and 5c, we can see that the predicted $\mu^*$ is a nonlinear function ($\mu^*$ does not vary monotonically with $T_a$ and RH) of ambient temperature and RH. For example, $\mu^*$ decreases first and then increases at $T_a$=16.5 °C. For various RH (8.5 %, 25.5 %, and 42.5 %), $\mu^*$ showed a similar pattern with increasing temperature: $\mu^*$ first decreased and then exhibited an upward trend.

(ii) As we discussed above, the large-scale electrification equilibrium is evidenced by the large squared wavelet coherence $R^2(n,s)$, as well as the straight line in the plots of ρ vs. $M_{10}$ in Fig. 4. Meanwhile, as we stated in the response of comment 38, the occasional absence of the large-scale electrification equilibrium is evidenced by a weak correlation between space charge density and dust concentration.

**Comment 41:** References: Suggest adding Williams et al. (2009) and studying it.

**Response:**

We have already studied the reference of Williams et al. (2009) in our previous work (for example, Zhang et al. 2017). According to the reviewer's suggestion, Williams et al. (2009) have also been cited in the revised manuscript. For example:

The area was selected since it lies within a dusty belt in the Hexi Corridor (Wang et al., 2018), which is mainly affected by the Mongolian cyclones (and probably by the cold downdrafts from thunderstorms/squall) during the observational period and is therefore frequently subjected to dust events (Shao, 2000; Williams et al., 2009).

Previous measurements have demonstrated that the charge structure of dust clouds in dust storms could appear as unipolar, bipolar, and even multipolar. For example, Williams et al. (2009) measured the vertical E-field in dust storms and found both upward- and downward-pointing vertical E-field. They inferred that the dust cloud is unipolar if the near-ground particle charge transfer is dominating, while the dust cloud is bipolar if upper-air (volume) charge transfer is dominating.